# Variational Autoencoders with Jointly Optimized Latent Dependency Structure

**Jiawei He**[1]* **& Yu Gong**[1]*
{jha203, gongyug}@sfu.ca

**Joseph Marino**[2]
jmarino@caltech.edu

**Greg Mori**[1]
mori@cs.sfu.ca

**Andreas M. Lehrmann**
andreas.lehrmann@gmail.com

[1]School of Computing Science, Simon Fraser University
 Burnaby, BC, V5B1Z1, Canada

[2]California Institute of Technology
 Pasadena, CA, 91125, USA

## ABSTRACT

We propose a method for learning the dependency structure between latent variables in deep latent variable models. Our general modeling and inference framework combines the complementary strengths of deep generative models and probabilistic graphical models. In particular, we express the latent variable space of a variational autoencoder (VAE) in terms of a Bayesian network with a learned, flexible dependency structure. The network parameters, variational parameters as well as the latent topology are optimized simultaneously with a single objective. Inference is formulated via a sampling procedure that produces expectations over latent variable structures and incorporates top-down and bottom-up reasoning over latent variable values. We validate our framework in extensive experiments on MNIST, Omniglot, and CIFAR-10. Comparisons to state-of-the-art structured variational autoencoder baselines show improvements in terms of the expressiveness of the learned model.

## 1 INTRODUCTION

Deep latent variable models offer an effective method for automatically learning structure from data. By explicitly modeling the data distribution using latent variables, these models are capable of learning compressed representations that are then relevant for downstream tasks. Such models have been applied across a wide array of domains, such as images (Gregor et al., 2014; Kingma & Welling, 2014; Rezende et al., 2014; Gregor et al., 2015), audio (Chung et al., 2015; Fraccaro et al., 2016), video (He et al., 2018; Yingzhen & Mandt, 2018), and text (Bowman et al., 2016; Krishnan et al., 2017). However, despite their success, latent variable models are often formulated with simple (e.g. Gaussian) distributions, making independence assumptions among the latent variables. That is, each latent variable is sampled independently. Ignoring the dependencies between latent variables limits the flexibility of these models, negatively impacting the model's ability to fit the data.

In general, structural dependencies can be incorporated into all phases of a forward process, including inference, latent model, and output space: Normalizing flows (Rezende & Mohamed, 2015), for instance, accounts for dependencies during inference by learning a mapping from a simple distribution to a more complex distribution that contains these dependencies. Structured output networks (Lehrmann & Sigal, 2017), on the other hand, directly predict an expressive non-parametric output distribution. On the modeling side, one can add dependencies by constructing a *hierarchical* latent representation (Dayan et al., 1995). These structures consist of conditional (*empirical*) priors, in which one latent variable forms a prior on another latent variable. While this conditional

---

*Equal Contribution.

distribution may take a simple form, marginalizing over the parent variable can result in an arbitrarily complex distribution. Models with these more flexible latent dependency structures have been shown to result in improved performance (Sønderby et al., 2016; Burda et al., 2016; Kingma et al., 2016). However, despite the benefits of including additional structure in these models, their dependency structures have so far been predefined, potentially limiting the performance of this approach.

In this work, we propose a method for learning dependency structures in latent variable models. Structure learning is a difficult task with a long history in the graphical models community (Koller & Friedman, 2009). Over the years, it has been tackled from several perspectives, including constraint-based approaches (Cheng et al., 2002; Lehmann & Romano, 2008), optimization of structure scores (Kass & Raftery, 1995; Heckerman et al., 1995; Barron et al., 1998), Bayesian model averaging (Heckerman et al., 1999; Koivisto & Sood, 2004), and many more. Unfortunately, the underlying objectives are often limited to graphs of a particular form (e.g., limited tree width), prohibitively expensive, or difficult to integrate with the gradient-based optimization techniques of modern neural networks. Here, we discuss an end-to-end approach for general graph structures introducing minimal complexity overhead. In particular, we introduce a set of binary global variables to gate the latent dependencies. The whole model (including its structure) is jointly optimized with a single stochastic variational inference objective. In our experimental validation, we show that the learned dependency structures result in models that more accurately model the data distribution, outperforming several common predefined latent dependency structures.

## 2  BACKGROUND

### 2.1  VARIATIONAL INFERENCE & VARIATIONAL AUTOENCODERS

A latent variable model, defined by the joint distribution, $p_\theta(\mathbf{x}, \mathbf{z}) = p_\theta(\mathbf{x}|\mathbf{z})p_\theta(\mathbf{z})$, models each data example, $\mathbf{x}$, using a local latent variable, $\mathbf{z}$, and global parameters, $\theta$. $p_\theta(\mathbf{x}|\mathbf{z})$ denotes the conditional likelihood, and $p_\theta(\mathbf{z})$ denotes the prior. Latent variable models are capable of capturing the structure present in data, with $\mathbf{z}$ forming a compressed representation of each data example. Unfortunately, inferring the posterior, $p_\theta(\mathbf{z}|\mathbf{x})$, is typically computationally intractable, prompting the use of approximate inference techniques. Variational inference (Jordan et al., 1999) introduces an approximate posterior, $q_\phi(\mathbf{z}|\mathbf{x})$, and optimizes variational parameters, $\phi$, to minimize the KL-divergence to the true posterior, $\mathrm{KL}(q_\phi(\mathbf{z}|\mathbf{x})\|p_\theta(\mathbf{z}|\mathbf{x}))$. As this quantity cannot be evaluated directly, the following relation is used:

$$\log p_\theta(\mathbf{x}) = \mathrm{KL}(q_\phi(\mathbf{z}|\mathbf{x})\|p_\theta(\mathbf{z}|\mathbf{x})) + \mathcal{L}(\mathbf{x}; \theta, \phi), \tag{1}$$

where $\mathcal{L}(\mathbf{x}; \theta, \phi)$ is the evidence lower bound (ELBO), defined as

$$\mathcal{L}(\mathbf{x}; \theta, \phi) = \mathbb{E}_{q_\phi(\mathbf{z}|\mathbf{x})} \left[\log p_\theta(\mathbf{x}|\mathbf{z})\right] - \mathrm{KL}(q_\phi(\mathbf{z}|\mathbf{x})\|p_\theta(\mathbf{z})). \tag{2}$$

In Eq. (1), $\log p_\theta(\mathbf{x})$ is independent of $\phi$, so we can minimize the KL divergence term, i.e. perform approximate inference, by maximizing $\mathcal{L}(\mathbf{x}; \theta, \phi)$ w.r.t. $q_\phi(\mathbf{z}|\mathbf{x})$. Further, because KL divergence is non-negative, $\mathcal{L}(\mathbf{x}; \theta, \phi)$ is a lower bound on $\log p_\theta(\mathbf{x})$, meaning we can then learn the model parameters by maximizing $\mathcal{L}(\mathbf{x}; \theta, \phi)$ w.r.t. $\theta$.

Variational autoencoders (VAEs) (Kingma & Welling, 2014; Rezende et al., 2014) amortize inference optimization across data examples by parameterizing $q_\phi(\mathbf{z}|\mathbf{x})$ as a separate inference model, then jointly optimizing the model parameters $\theta$ and $\phi$. VAEs instantiate both the inference model and latent variable model with deep networks, allowing them to scale to high-dimensional data. However, VAEs are typically implemented with basic graphical structures and simple, unimodal distributions (e.g. Gaussians). For instance, the dimensions of the prior are often assumed to be independent, $p_\theta(\mathbf{z}) = \prod_m p_\theta(z_m)$, with a common assumption being a fixed standard Gaussian: $p_\theta(\mathbf{z}) = \mathcal{N}(\mathbf{z}; \mathbf{0}, \mathbf{I})$. Similarly, approximate posteriors often make the mean field assumption, $q_\phi(\mathbf{z}|\mathbf{x}) = \prod_m q_\phi(z_m|\mathbf{x})$. Independence assumptions such as these may be overly restrictive, thereby limiting modeling capabilities.

### 2.2  EMPIRICAL PRIORS THROUGH LATENT DEPENDENCY STRUCTURE

One technique for improving the expressive capacity of latent variable models is through incorporating dependency structure among the latent variables, forming a hierarchy (Dayan et al., 1995;

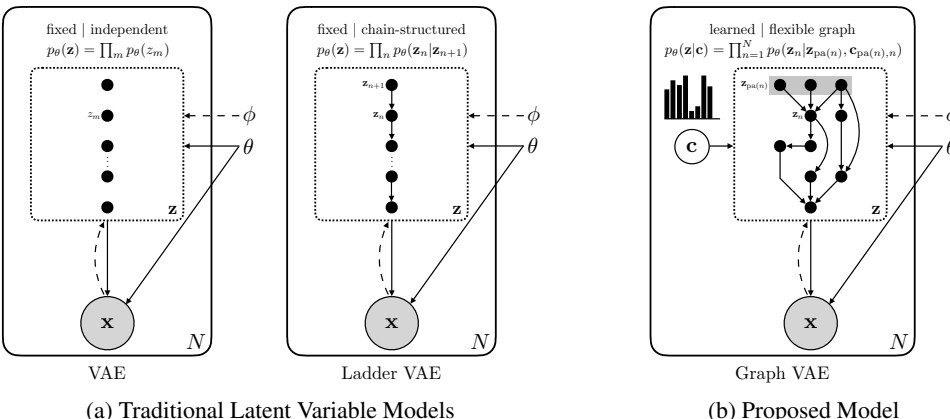

(a) Traditional Latent Variable Models          (b) Proposed Model

Figure 1: **Overview: Model Comparison.** We show the graphical representations of **(a)** traditional latent variable models (VAE, ladder VAE) and **(b)** the proposed graph VAE. Solid lines denote generation, dashed lines denote inference, and the dotted area indicates the latent space governed by variational parameters $\phi$ and generative parameters $\theta$. Both VAE and ladder VAE use a fixed graph structure with limited expressiveness (VAE: independent; ladder VAE: chain-structured). In contrast, graph VAE jointly optimizes a distribution over latent structures $\mathbf{c}$ and model parameters $(\phi, \theta)$, allowing test-time sampling of a flexible, data-driven latent structure.

Rezende et al., 2014; Goyal et al., 2017; Vikram et al., 2018; Webb et al., 2018). These dependencies provide *empirical* priors, learned priors that are conditioned on other latent variables. With $M$ latent dimensions, the full prior takes the following auto-regressive form:

$$p_\theta(\mathbf{z}) = \prod_{m=1}^{M} p_\theta\big(z_m | \mathbf{z}_{\mathrm{pa}(m)}\big), \tag{3}$$

where $\mathbf{z}_{\mathrm{pa}(m)}$ denotes the vector of latent variables constituting the parents of $z_m$. Each conditional distribution can be parameterized by deep networks that output the parameters of distributions, e.g. mean and variance of a Gaussian. While these conditional distributions may be relatively simple, the marginal empirical prior, $p_\theta(z_m) = \int p_\theta(z_m | \mathbf{z}_{\mathrm{pa}(m)}) p_\theta(\mathbf{z}_{\mathrm{pa}(m)}) \mathrm{d}\mathbf{z}_{\mathrm{pa}(m)}$, can be arbitrarily complex. By using more flexible priors, models with latent dependency structure are less restrictive in their latent representations, hopefully capturing more information and enabling a better fit to the data.

With the added latent dependencies in the model, the independence assumption in the approximate posterior is even less valid in this setting. While normalizing flows (Rezende & Mohamed, 2015) offers one technique for overcoming the mean field assumption, a separate line of work has investigated the use of *structured* approximate posteriors, particularly in the context of models with empirical priors (Johnson et al., 2016). This technique introduces dependencies between the dimensions of the approximate posterior, often mirroring the dependency structure of the latent variable model. An explanation for this was provided by Marino et al. (2018): optimizing the approximate posterior requires knowledge of the prior, which is especially relevant in models with empirical priors where the prior can vary with the data. Ladder VAE (Sønderby et al., 2016) incorporates these prior dependencies by using a structured approximate posterior of the form

$$q_\phi(\mathbf{z}|\mathbf{x}) = \prod_{m=1}^{M} q_\phi\big(z_m | \mathbf{x}, \mathbf{z}_{\mathrm{pa}(m)}\big). \tag{4}$$

Unlike the mean field approximate posterior, which conditions each dimension only on the data example, $\mathbf{x}$, the distributions in Eq. (4) account for latent dependencies by conditioning on samples from the parent variables. Ladder VAE performs this conditioning by reusing the empirical prior during inference, forming the approximate posterior by combining a "bottom-up" recognition distribution and the "top-down" prior.

While Eqs. (3) and (4) permit separate latent dependencies for each individual latent dimension, the dimensions are typically partitioned into a set of nodes, with dimensions within each node sharing

the same parents. This improves computational efficiency by allowing priors and approximate posteriors within each node to be calculated in parallel. Using $\mathbf{z}_n$ to denote latent node $n$ of $N$ and $\mathbf{z}_{\text{pa}(n)}$ to denote the concatenation of its parent nodes, we can write the ELBO (Eq. (2)) as

$$\mathcal{L}(\mathbf{x}; \theta, \phi) = \mathbb{E}_{q_\phi(\mathbf{z}|\mathbf{x})} \left[ \log p_\theta(\mathbf{x}|\mathbf{z}) \right] - \sum_{n=1}^{N} \mathbb{E}_{q_\phi(\mathbf{z}|\mathbf{x})} \left[ \log \frac{q_\phi(\mathbf{z}_n|\mathbf{x}, \mathbf{z}_{\text{pa}(n)})}{p_\theta(\mathbf{z}_n|\mathbf{z}_{\text{pa}(n)})} \right]. \tag{5}$$

Note that the KL divergence term in the ELBO can no longer be evaluated analytically, now requiring a sampling-based estimate of the expectation (Kingma & Welling, 2014). While this can lead to higher variance in ELBO estimates and the resulting gradients, models with latent dependencies still tend to empirically outperform models with independence assumptions (Burda et al., 2016; Sønderby et al., 2016). However, by increasing the number of nodes, the burden of devising a suitable dependency structure falls upon the experimental practitioner. This is non-trivial, as the structure may depend on the data and other model hyperparameters, such as the number of layers in the deep networks, non-linearities, latent distributions, etc. Rather than relying on pre-defined fully-connected structures (Kingma et al., 2016) or chain structures (Sønderby et al., 2016), we seek to automatically learn the latent dependency structure as part of the variational optimization process. A comparison of these approaches is visualized in Fig. 1.

## 3 Variational Optimization of Latent Structures

Unlike the model parameters $(\phi, \theta)$, which are optimized over a continuous domain, the latent dependency structure is *discrete*, without a clear ordering. The discrete nature of the latent space's topological structure introduces discontinuities in the optimization landscape, complicating the learning process. Fortunately, unlike the related setting of neural architecture search (Zoph & Le, 2016), there is only a finite number of possible dependency structures over a fixed number of latent dimensions: In a directed graphical model, a fully-connected directed acyclic graph (DAG) models all possible dependencies. In this model, an ordering is induced over the latent nodes, and the parents of node $n$ (of $N$) are given as $\mathbf{z}_{\text{pa}(n)} = \{\mathbf{z}_{n+1}, \ldots, \mathbf{z}_N\}$.[1] Thus, to learn an appropriate latent dependency structure, we can maintain all dependencies in a fully-connected DAG, modifying their presence or absence during training. This is accomplished by introducing a set of binary dependency *gates* (Section 3.1). We convert discrete optimization over dependency structures into a continuous optimization problem by parameterizing these gates as samples from Bernoulli distributions, then learning the distribution parameters (Section 3.3). These gating distributions induce an additional lower bound on $\mathcal{L}$, which becomes tight when the distribution converges to a delta function, yielding a single, optimized dependency structure (Section 3.2). Indeed, we observe this process empirically, with the learned dependency structures outperforming their predefined counterparts (Section 4).

### 3.1 Gated Dependencies

To control the dependency structure of the model, we introduce a set of binary global variables, $\mathbf{c} = \{c_{i,j}\}_{i,j}$, which *gate* the latent dependencies. The element $c_{i,j}$ denotes the gate variable from $\mathbf{z}_i$ to $\mathbf{z}_j$ $(i > j)$, specifying the presence or absence of this latent dependency. Because each element of $\mathbf{c}$ takes values in $\{0, 1\}$, dependencies can be preserved or removed simply through (broadcasted) element-wise multiplication with the corresponding parent nodes. Removing a dependency entails multiplying the corresponding input parent node by $0$. Each possible latent dependency structure can now be expressed through its corresponding value of $\mathbf{c}$.

Each fixed latent dependency structure, $\mathbf{c}'$, induces a separate latent variable model $p_\theta(\mathbf{x}, \mathbf{z}, \mathbf{c}) = p_\theta(\mathbf{x}|\mathbf{z}, \mathbf{c}) p_\theta(\mathbf{z}|\mathbf{c}) \delta_{\mathbf{c}, \mathbf{c}'}$, where $\delta_{\cdot,\cdot}$ is the Kronecker delta, which effectively selects a single structure. Similar to Eq. (3), the prior on the latent variables can now be expressed as

$$p_\theta(\mathbf{z}|\mathbf{c}) = \prod_{n=1}^{N} p_\theta(\mathbf{z}_n|\mathbf{z}_{\text{pa}(n)}, \mathbf{c}_{\text{pa}(n),n}), \tag{6}$$

where $\mathbf{c}_{\text{pa}(n),n}$ denotes the gate variables associated with the dependencies between node $\mathbf{z}_n$ and its parents, $\mathbf{z}_{\text{pa}(n)}$. Note that $\mathbf{z}_{\text{pa}(n)}$ denotes the set of all *possible* parents of node $\mathbf{z}_n$ in the fully-connected DAG, i.e. $\mathbf{z}_{\text{pa}(n)} = \{\mathbf{z}_{n+1}, \ldots, \mathbf{z}_N\}$. To give a concrete example of the gating procedure,

---

[1]We follow convention (Dayan et al., 1995; Rezende et al., 2014; Sønderby et al., 2016), with parent nodes having a larger index than their children.

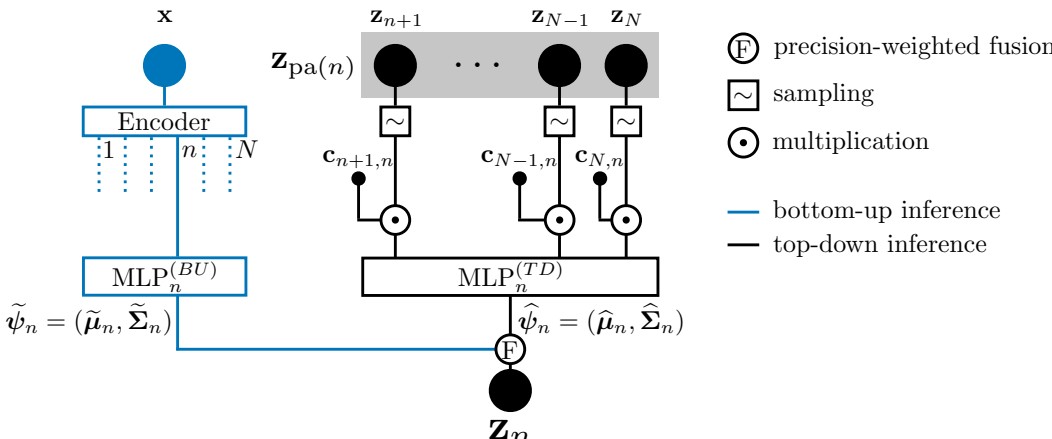

Figure 2: **Local Distributions.** We illustrate the parametrization of a local variable $\mathbf{z}_n$ in our structured representation. The local prior (Eq. (6)) is defined in terms of a top-down process (in black) predicting the node's parameters $\widehat{\psi}_n$ from a gate-modulated sample of $\mathbf{z}_n$'s parents $\mathbf{z}_{\mathrm{pa}(n)}$. The local approximate posterior (Eq. (7)) additionally performs a precision-weighted fusion of these parameters with the result of a bottom-up process using a node-specific MLP to predict input-conditioned parameters $\widetilde{\psi}_n$ from a generic encoding of $\mathbf{x}$.

consider the case in which $p_\theta(\mathbf{z}_n|\mathbf{z}_{\mathrm{pa}(n)}, \mathbf{c}_{\mathrm{pa}(n),n})$ is given by a Gaussian density with parameters $\widehat{\psi}_n = (\widehat{\boldsymbol{\mu}}_n, \widehat{\boldsymbol{\Sigma}}_n)$. We obtain these parameters recursively by multiplying samples of node $\mathbf{z}_n$'s parent variables $\mathbf{z}_{\mathrm{pa(n)}}$ with their corresponding gating variables $\boldsymbol{c}_{\mathrm{pa(n),n}}$ and input a concatenation of the results of this operation into a multi-layer perceptron $\mathrm{MLP}_n^{(TD)}$ predicting $\widehat{\psi}_n$ (see Appendix B for additional details on the MLP architecture). The top-down recursion starts at the root node $\mathbf{z}_N \sim p(\mathbf{z}_N) = \mathcal{N}(\mathbf{0}, \mathbf{I})$. An illustration of this process is shown in black in Fig. 2.

The approximate posterior, $q_\phi(\mathbf{z}|\mathbf{x}, \mathbf{c})$, must approximate $p_\theta(\mathbf{z}|\mathbf{x}, \mathbf{c})$. We express the approximate posterior as

$$q_\phi(\mathbf{z}|\mathbf{x}, \mathbf{c}) = \prod_{n=1}^{N} q_\phi(\mathbf{z}_n|\mathbf{x}, \mathbf{z}_{\mathrm{pa}(n)}, \mathbf{c}_{\mathrm{pa}(n)}). \tag{7}$$

We note that the dependency structures of the generative model and its corresponding posterior are, in general, not independent: choosing a particular structure in the generative model induces a particular structure in the posterior (Webb et al., 2018). A simple way to guarantee enough capacity in the encoder to account for the dependencies implied by the decoder is thus to keep the encoder graph fully-connected and learn a decoder graph only. Instead, we *share* the gating variables $\mathbf{c}$ between approximate posterior and generative model (see Section 3.3), i.e., we assume that the encoder dependencies mirror those of the decoder. As a consequence, the posterior implied by the generative model could lie outside of the model class representable by the encoder. In practice, this is not an issue and we observe significant performance improvements over both traditional VAEs (where prior and posterior match but are limited in their expressiveness) and Graph VAEs with fully-connected encoder graph. See Section 4.3 for quantitative experiments and Section 5 for a discussion on the relationship between learned structures and fully-connected structures.

Parameter prediction for the local factors $q_\phi(\mathbf{z}_n|\mathbf{x}, \mathbf{z}_{\mathrm{pa}(n)}, \mathbf{c}_{\mathrm{pa}(n)})$ consists of a precision-weighted fusion of the top-down prediction $\widehat{\psi}_n$ described above and a bottom-up prediction $\widetilde{\psi}_n$. The latter is obtained by encoding $\mathbf{x}$ into a generic feature that is used as an input to a node-specific multi-layer perceptron $\mathrm{MLP}_n^{(BU)}$ predicting $\widetilde{\psi}_n$. This is shown in blue in Fig. 2. Additional details on the fusion process can be found in Appendix B.3.

---

**Algorithm 1** Optimizing VAEs with Latent Dependency Structure

---

**Require:** Data $\mathbf{x}$, number of latent nodes $N$, number of dimensions per node $N'$.
 1: Initialize $\theta, \phi, \boldsymbol{\mu}$.
 2: **repeat**
 3:      Sample $\mathbf{c}$ using Eq. (9) and determine $\mathbf{z}_{\mathrm{pa}(n)}$ for each $\mathbf{z}_n$ based on the sampled structure.
 4:      For each node, compute $q_\phi(\mathbf{z}_n|\mathbf{x}, \mathbf{z}_{\mathrm{pa}(n)})$ using Eq. (7).
 5:      Sample $\mathbf{z}$ from $q_\phi(\mathbf{z}|\mathbf{x})$ using Eq. (6) and compute $p_\theta(\mathbf{x}|\mathbf{z})$.
 6:      Update $\theta, \phi, \boldsymbol{\mu}$ based on the gradients derived from Eq. (8).
 7: **until** Convergence.

---

## 3.2 LEARNING STRUCTURE BY INDUCING AN ADDITIONAL LOWER BOUND

The formulation in Section 3.1 provides the form of the model for a particular configuration of the latent dependency structure. Finding the optimal structure corresponds to a discrete optimization over all values of $\mathbf{c}$, potentially optimizing the model parameters of each possible configuration. To avoid this intractable procedure, we place a distribution over $\mathbf{c}$, then directly optimize the parameters of this distribution to arrive at a single, learned latent dependency structure. Specifically, we treat each $c_{i,j}$ as an independent random variable, sampled from a Bernoulli distribution with mean $\mu_{i,j}$, i.e. $c_{i,j} \sim p(c_{i,j}) = \mathcal{B}(\mu_{i,j})$. We denote the set of these Bernoulli means as $\boldsymbol{\mu}$. Introducing this distribution allows us to express the following additional lower bound on $\mathcal{L}$, derived in Appendix A:

$$\mathcal{L} \geq \widetilde{\mathcal{L}} = \mathbb{E}_{p(\mathbf{c})}\left[\mathbb{E}_{q_\phi(\mathbf{z}|\mathbf{x})}\left[\log p_\theta(\mathbf{x}|\mathbf{z}, \mathbf{c})\right] - \mathrm{KL}(q_\phi(\mathbf{z}|\mathbf{x})\|p_\theta(\mathbf{z}|\mathbf{c}))\right] = \mathbb{E}_{p(\mathbf{c})}\left[\mathcal{L}_\mathbf{c}\right], \tag{8}$$

where $\mathcal{L}_\mathbf{c}$ is the ELBO for a particular value of dependency gating variables. Thus, $\widetilde{\mathcal{L}}$ can be interpreted as the expected ELBO under the distribution of dependency structures induced by $p(\mathbf{c})$, which we estimate by sampling $\mathbf{c} \sim p(\mathbf{c})$ and evaluating $\mathcal{L}_\mathbf{c}$. We note that $\widetilde{\mathcal{L}}$ is not a proper variational bound, as it is not guaranteed to recover the marginal likelihood if the approximate posterior matches the true posterior. Rather, optimizing $\widetilde{\mathcal{L}}$ provides a method for learning the latent structure. For fixed parameters $(\phi, \theta)$, the optimal $\widetilde{\mathcal{L}}$ w.r.t. $\boldsymbol{\mu}$ is a $\delta$-distribution at the MAP configuration, $\mathbf{c}^*$, yielding $\mathcal{L} = \widetilde{\mathcal{L}} = \mathcal{L}_{\mathbf{c}^*}$. This is because $\mathcal{L}_{\mathbf{c}^*}$ is always greater than or equal to the expected ELBO over all dependency gates, $\widetilde{\mathcal{L}}$. In practice, we jointly optimize $\phi$, $\theta$, and $\boldsymbol{\mu}$. While this non-convex optimization procedure may result in local optima, we find this empirically works well, with $p(\mathbf{c})$ converging to a fixed distribution (Fig. 3). Thus, by the end of training, we are effectively optimizing the ELBO for a single dependency structure. The training procedure is outlined in Algorithm 1.

## 3.3 LEARNING THE DISCRETE GATING VARIABLE DISTRIBUTIONS

For a given latent dependency structure, gradients for the parameters $\theta$ and $\phi$ can be estimated using Monte Carlo samples and the reparameterization trick (Kingma & Welling, 2014; Rezende et al., 2014). To obtain gradients for the gate means, $\boldsymbol{\mu}$, we make use of recent advances in differentiating through discrete operations (Maddison et al., 2017; Jang et al., 2017), allowing us to differentiate through the sampling of the dependency gating variables, $\mathbf{c}$. Specifically, we recast the gating variables using the Gumbel-Softmax estimator from Jang et al. (2017), re-expressing $c_{i,j}$ as:

$$c_{i,j} = \frac{\exp((\log(\mu_{i,j}) + \epsilon_1)/\tau)}{\exp((\log(\mu_{i,j}) + \epsilon_1)/\tau) + \exp((\log(1 - \mu_{i,j}) + \epsilon_2)/\tau)}, \tag{9}$$

where $\epsilon_1$ and $\epsilon_2$ are i.i.d samples drawn from a Gumbel(0, 1) distribution and $\tau$ is a temperature parameter. The Gumbel-Softmax distribution is differentiable for $\tau > 0$, allowing us to estimate the derivative $\frac{\partial c_{i,j}}{\partial \mu_{i,j}}$. For large values of $\tau$, we obtain smoothed versions of $\mathbf{c}$, essentially interpolating between different dependency structures. As $\tau \to 0$, we recover binary values for $\mathbf{c}$, yielding the desired discrete sampling of dependency structures at the cost of high-variance gradient estimates. Thus, we anneal $\tau$ during training to learn the dependency gate means, $\boldsymbol{\mu}$, eventually arriving at the discrete setting.

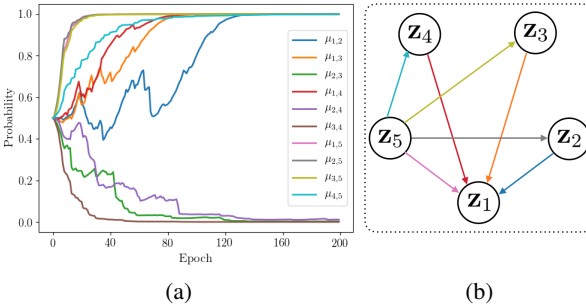 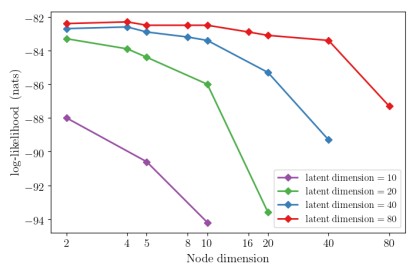

(a)        (b)

Figure 3: **Structure Learning.** In (**a**), we show training of the Bernoulli parameters $\mu_{i,j}$ governing the distribution over graph structures in architecture space. All edges are color-coded and can be located in (**b**), where we show a random sample from the resulting steady-state distribution with the same color scheme.

Figure 4: **Ablation Study on MNIST.** For a fixed latent dimension $M$ (color-coded), we report the log-likelihood of all possible factorizations of node dimension $N'$ ($x$-axis) and number of nodes $N = M/N'$.

## 4 EVALUATION

We evaluate the proposed latent dependency learning approach on three benchmark datasets: MNIST (Lecun et al., 1998; Larochelle & Murray, 2011), Omniglot (Lake et al., 2013), and CIFAR-10 (Krizhevsky, 2009). After discussing the experimental setup in Section 4.1, we provide a set of qualitative experiments in Section 4.2 to gain insight into the learning process, hyper-parameter selection, and the nature of the inferred structures. In Section 4.3, we provide quantitative comparisons with common predefined latent dependency structures on benchmark datasets. Additional results on training robustness and latent space embeddings can be found in Appendix C.

### 4.1 EXPERIMENTAL SETUP

To provide a fair comparison, the encoders of all structured methods use the same MLP architecture with batch normalization (Ioffe & Szegedy, 2015) and ReLU non-linearities (Nair & Hinton, 2010) in all experiments.[2] Decoder structures are the reverse of the encoders. Likewise, the number of latent dimensions is the same in all models and experiments ($M = 80$). As discussed in Section 3.1, all latent dependencies are modeled by non-linear MLPs as well.

For MNIST and Omniglot, we binarize the data and model $p_\theta(\mathbf{x}|\mathbf{z})$ as a Bernoulli distribution, using a sigmoid non-linearity on the output layer to produce the mean of this distribution. For CIFAR-10, we model $p_\theta(\mathbf{x}|\mathbf{z})$ with a Gaussian density, with mean and log-variance predicted by sigmoid and linear functions, respectively. Further implementation details, including the model architectures and training criteria, can be found in Appendix B.

### 4.2 QUALITATIVE ANALYSIS

We first explore the structure learning process. As described in Section 3.2 and Appendix A, optimizing $\widetilde{\mathcal{L}}$ w.r.t. the dependency gating means, $\boldsymbol{\mu}$, should push this lower bound toward $\mathcal{L}$. Thus, $\boldsymbol{\mu}$ should converge to either 0 or 1, yielding a fixed, final latent dependency structure. In Fig. 3, we visualize this process during training on MNIST. The model has $N = 5$ nodes and a total latent dimension of $M = 80$ (i.e., $N' = M/N = 16$ dimensions per node). As shown in Fig. 3a, the gating means converge in practice, with 3 out of 10 edges removed and the rest retained. The resulting static dependency structure is visualized in Fig. 3b. We observed that the learned structure is stable across training runs with different seeds for parameter initialization and mini-batch sampling, supporting the hypothesis that the inferred structure indeed depends on the model parameterization and the dataset.

---

[2]We implement classic VAEs using a more complex encoder to match the number of parameters of the structured methods. All baselines use the same or more parameters than Graph VAE.

| Dataset | Method | LL | KL | ELBO |
|---|---|---|---|---|
| MNIST | VAE | $-87.30 \pm 0.04$ | $26.28 \pm 0.05$ | $-95.61 \pm 0.07$ |
| | Ladder VAE | $-84.98 \pm 0.10$ | $24.25 \pm 0.12$ | $-92.80 \pm 0.14$ |
| | FC-VAE | $-82.80 \pm 0.15$ | $\mathbf{22.47 \pm 0.09}$ | $-88.49 \pm 0.12$ |
| | Graph VAE | $\mathbf{-82.58 \pm 0.08}$ | $22.49 \pm 0.10$ | $\mathbf{-88.35 \pm 0.14}$ |
| Omniglot | VAE | $-109.62 \pm 0.09$ | $32.11 \pm 0.07$ | $-115.43 \pm 0.12$ |
| | Ladder VAE | $-105.92 \pm 0.16$ | $\mathbf{29.46 \pm 0.10}$ | $-110.62 \pm 0.17$ |
| | FC-VAE | $-105.01 \pm 0.26$ | $30.74 \pm 0.12$ | $-108.79 \pm 0.21$ |
| | Graph VAE | $\mathbf{-104.57 \pm 0.19}$ | $29.93 \pm 0.10$ | $\mathbf{-108.17 \pm 0.24}$ |
| CIFAR-10 | VAE | $-6.37 \pm \triangle$ | $0.10 \pm \triangle$ | $-6.42 \pm \triangle$ |
| | Ladder VAE | $-6.10 \pm 0.02$ | $0.08 \pm \triangle$ | $-6.13 \pm 0.02$ |
| | FC-VAE | $-5.89 \pm 0.04$ | $\mathbf{0.07} \pm \triangle$ | $-5.90 \pm 0.04$ |
| | Graph VAE | $\mathbf{-5.81 \pm 0.04}$ | $\mathbf{0.07} \pm \triangle$ | $\mathbf{-5.82 \pm 0.04}$ |

Table 1: **Quantitative Analysis.** Test log-likelihood (LL), $D_{\mathrm{KL}}(q_\phi(\mathbf{z}|\mathbf{x})\|p_\theta(\mathbf{z}))$ (KL), and ELBO for the proposed Graph VAE model and the baseline models with predefined dependency structures. We show mean and standard deviation over 5 independent runs, where $\triangle$ indicates a value $< 0.01$.

We next investigate the influence of the total latent dimension, $M$, and the trade-off between the number of nodes, $N$, and the node dimension, $N' = M/N$. Our results for various models trained on MNIST are shown in Fig. 4. Models with the same total latent dimension are shown in the same color. We observe that the performance improves with increasing total latent dimension, likely resulting from the additional flexibility of the higher-dimensional latent space. We also observe that, for a fixed number of latent dimensions, models with fewer node dimensions (and therefore more nodes with a more complex dependency structure) typically perform better. This highlights the importance of using an expressive dependency structure for obtaining a flexible model.

### 4.3 QUANTITATIVE COMPARISON

To quantitatively evaluate the improvements due to learning the latent dependency structure, we compare with a range of common, predefined baseline structures. These baselines include classic VAEs (Kingma & Welling, 2014; Rezende et al., 2014), which contain no dependencies in the prior, Ladder VAEs (Sønderby et al., 2016), which contain chain-like dependencies in the prior, and fully-connected VAEs (FC-VAEs) (cf. Kingma et al. (2016)), which contain all possible dependencies in the prior (corresponding to all gating variable parameters $\mu_{i,j}$ set to a fixed value of 1). We note that our approach is orthogonal and could be complemented by a number of other approaches attempting to overcome the limitations of classic VAEs. Similar to ladder VAEs, Zhao et al. (2017) use chain-structured latent dependencies to learn disentangled representations. Normalizing flows (Rezende & Mohamed, 2015), on the other hand, adds dependencies to the approximate posterior through a series of invertible transformations.

We evaluate the performance of all models using their test log-likelihood, $\log p_\theta(\mathbf{x})$, in 5 independent runs (Table 1). All values were estimated using $5,000$ importance-weighted samples. Following standard practice, we report $\log p_\theta(\mathbf{x})$ in *nats* on MNIST/Omniglot and in *bits/input dimension* on CIFAR-10. The learned dependency structure in our proposed Graph VAE consistently outperforms models with both fewer (VAE, ladder VAE) and more (FC-VAE) latent dependencies. We discuss potential reasons in Section 5. To provide further insight into the training objective, Table 1 also reports $D_{\mathrm{KL}}(q_\phi(\mathbf{z}|\mathbf{x})\|p_\theta(\mathbf{z}))$ and the ELBO for each model on the test set.

**Encoder-Decoder Relationship.** From a purely theoretical point of view, learning the structure of the generative model implies the need for a fully-connected graph in the approximate posterior (see Section 3.1). In practice, we share the gating variables $\mathbf{c}$ between encoder and decoder (see Eq. (8)), because we observed improved empirical performance when doing so: training a Graph VAE decoder with a predefined, fully-connected encoder graph results in a mean test log-likelihood of $-83.40$ nats on MNIST, which is worse than the performance of FC-VAE ($-82.80$ nats) and Graph VAE ($-82.58$ nats). We believe these are noteworthy empirical results, but further research is

required to understand this behavior at a theoretical level. A full visualization of the training process is provided in Appendix C.1.

## 5 DISCUSSION

**Performance vs. Speed/Memory.**    As shown in Fig. 4, the number of latent nodes can significantly impact the performance of our model. While allowing more complex dependency structures through a low $M/N$-ratio is typically beneficial, it also has an adverse effect on the training time and memory consumption. Fortunately, the ability to freely select this ratio allows a simple adaption to the available processing power, hardware constraints, and application scenarios.

**Optimization Order.**    It is worth noting that the learning process optimizes the model parameters $(\mathbf{c}, \phi, \theta)$ in a clear temporal order. While the latent structure, governed by $\mathbf{c}$, converges during the first $\approx 200$ epochs (Fig. 3a), it takes over $10\times$ as long until the variational and generative parameters $(\phi, \theta)$ converge. There is no external force enforcing this behaviour, indicating that the loss can initially most easily be decreased by limiting the latent structure to the complexity prescribed by the observed training data.

**Performance Improvement over Fully-Connected VAEs.**    There is an intricate relationship between fully-connected graphs vs. learned graphs along one axis and between prior structures vs. posterior structures along a separate axis: FC-VAEs model all conditional latent dependencies and are thus potentially more expressive and flexible than other latent dependency structures. It is therefore somewhat surprising that the learned latent structures in Graph VAE consistently outperform the FC-VAE baseline. We speculate that this may be due to difficulties in optimization, which is a known problem in hierarchical latent variable models (Bowman et al., 2016; Burda et al., 2016). Graph VAE and FC-VAE both contain the same hypothesis space of possible models that can be learned. If all dependencies are needed, Graph VAE could set all dependency gate parameters to 1. Likewise, if a latent dependency was unnecessary, FC-VAE could set all of the model parameters in that dependency to 0. However, this would require many coordinated steps along multiple parameter dimensions. It is plausible that the benefit of learning the dependency structure may stem from the ability to alter the optimization landscape, using the dependency gates to move through the model parameter space more coarsely and rapidly. The resulting latent dependency structures may thus be less expressive, but easier to optimize. While only an intuition, this hypothesis is also in line with the observations and results in our experiments with fully-connected encoder graphs and learned decoder graphs, where the theoretically more flexible FC-encoder is outperformed by our parameter sharing approach. Follow-up work will be required to test this intuition.

## 6 CONCLUSION

We presented a novel method for structure learning in latent variable models, which uses dependency gating variables, together with a modified objective and discrete differentiation techniques, to effectively transform discrete structure learning into a continuous optimization problem. In our experiments, the learned latent dependency structures improve the performance of latent variable models over comparable baselines with predefined dependency structures. The approach presented here provides directions for further research in structure learning for other tasks, including undirected graphical models, time-series models, and discriminative models.

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

# A    LOWER BOUND DERIVATION

Introducing the distribution over dependency gating variables, $p(\mathbf{c}) = \mathcal{B}(\boldsymbol{\mu})$, modifies the evidence lower bound (ELBO) from Eq. (2), as we now have an additional set of random variables over which to marginalize. To see this, we can start by re-expressing Eq. (2) as

$$\mathcal{L} = \mathbb{E}_{q_\phi(\mathbf{z}|\mathbf{x})}\left[\log p_\theta(\mathbf{x}, \mathbf{z})\right] - \mathbb{E}_{q_\phi(\mathbf{z}|\mathbf{x})}\left[\log q_\phi(\mathbf{z}|\mathbf{x})\right]. \tag{10}$$

$p_\theta(\mathbf{x}, \mathbf{z})$ can be expressed as a marginalization over the gating variables, effectively averaging over an ensemble of models with a distribution of dependency structures:

$$p_\theta(\mathbf{x}, \mathbf{z}) = \int p_\theta(\mathbf{x}, \mathbf{z}, \mathbf{c})d\mathbf{c} = \int p_\theta(\mathbf{x}, \mathbf{z}|\mathbf{c})p(\mathbf{c})d\mathbf{c} = \mathbb{E}_{p(\mathbf{c})}\left[p_\theta(\mathbf{x}, \mathbf{z}|\mathbf{c})\right]. \tag{11}$$

Plugging this into Eq. (10):

$$\mathcal{L} = \mathbb{E}_{q_\phi(\mathbf{z}|\mathbf{x})}\left[\log \mathbb{E}_{p(\mathbf{c})}\left[p_\theta(\mathbf{x}, \mathbf{z}|\mathbf{c})\right]\right] - \mathbb{E}_{q_\phi(\mathbf{z}|\mathbf{x})}\left[\log q_\phi(\mathbf{z}|\mathbf{x})\right]. \tag{12}$$

Using Jensen's inequality, we bring the $\log$ inside of the expectation, $\mathbb{E}_{p(\mathbf{c})}\left[\cdot\right]$, yielding

$$\mathcal{L} \geq \widetilde{\mathcal{L}} = \mathbb{E}_{q_\phi(\mathbf{z}|\mathbf{x})}\left[\mathbb{E}_{p(\mathbf{c})}\left[\log p_\theta(\mathbf{x}, \mathbf{z}|\mathbf{c})\right]\right] - \mathbb{E}_{q_\phi(\mathbf{z}|\mathbf{x})}\left[\log q_\phi(\mathbf{z}|\mathbf{x})\right], \tag{13}$$

where $\widetilde{\mathcal{L}}$ is a lower bound on $\mathcal{L}$. Swapping the order of expectation, we rewrite $\widetilde{\mathcal{L}}$ as

$$\widetilde{\mathcal{L}} = \mathbb{E}_{p(\mathbf{c})}\left[\mathbb{E}_{q_\phi(\mathbf{z}|\mathbf{x})}\left[\log p_\theta(\mathbf{x}, \mathbf{z}|\mathbf{c})\right]\right] - \mathbb{E}_{q_\phi(\mathbf{z}|\mathbf{x})}\left[\log q_\phi(\mathbf{z}|\mathbf{x})\right], \tag{14}$$

and because the second term is independent of $p(\mathbf{c})$, we can include both terms inside of a single outer expectation:

$$\begin{aligned}
\widetilde{\mathcal{L}} &= \mathbb{E}_{p(\mathbf{c})}\left[\mathbb{E}_{q_\phi(\mathbf{z}|\mathbf{x})}\left[\log p_\theta(\mathbf{x}, \mathbf{z}|\mathbf{c}) - \log q_\phi(\mathbf{z}|\mathbf{x})\right]\right] \\
&= \mathbb{E}_{p(\mathbf{c})}\left[\mathbb{E}_{q_\phi(\mathbf{z}|\mathbf{x})}\left[\log p_\theta(\mathbf{x}|\mathbf{z}, \mathbf{c})\right] - \mathrm{KL}(q_\phi(\mathbf{z}|\mathbf{x})\|p_\theta(\mathbf{z}|\mathbf{c}))\right] \\
&= \mathbb{E}_{p(\mathbf{c})}\left[\mathcal{L}_\mathbf{c}\right],
\end{aligned} \tag{15}$$

where we have defined $\mathcal{L}_\mathbf{c}$ as the ELBO for a given dependency structure. Note that $\widetilde{\mathcal{L}}$ is not a proper variational bound, as it cannot recover the marginal log likelihood. Rather, $\widetilde{\mathcal{L}}$ allows us to optimize the distribution over gating variables and, thus, the model structure. When we arrive at a fixed structure, we will be optimizing the variational bound for that particular dependency structure.

To see this, note that the bound in Eq. 13 becomes tight when $p(\mathbf{c})$ is any fixed distribution, in which the dependency gating means, $\boldsymbol{\mu}$, are all either $1$ or $0$. Plugging in a delta distribution for $p(\mathbf{c})$ at a particular configuration, $\mathbf{c}'$, i.e. $p(\mathbf{c}) = \delta_{\mathbf{c}, \mathbf{c}'}$, we have

$$\begin{aligned}
\mathcal{L} &= \mathbb{E}_{q_\phi(\mathbf{z}|\mathbf{x})}\left[\log \mathbb{E}_{\delta_{\mathbf{c}, \mathbf{c}'}}\left[p_\theta(\mathbf{x}, \mathbf{z}|\mathbf{c})\right]\right] - \mathbb{E}_{q_\phi(\mathbf{z}|\mathbf{x})}\left[\log q_\phi(\mathbf{z}|\mathbf{x})\right] \\
&= \mathbb{E}_{q_\phi(\mathbf{z}|\mathbf{x})}\left[\mathbb{E}_{\delta_{\mathbf{c}, \mathbf{c}'}}\left[\log p_\theta(\mathbf{x}, \mathbf{z}|\mathbf{c})\right]\right] - \mathbb{E}_{q_\phi(\mathbf{z}|\mathbf{x})}\left[\log q_\phi(\mathbf{z}|\mathbf{x})\right] \\
&= \widetilde{\mathcal{L}}.
\end{aligned} \tag{16}$$

Intuitively, this is simply the case of a latent variable model with predefined, fixed dependencies. By optimizing $\widetilde{\mathcal{L}}$ w.r.t. $\boldsymbol{\mu}$, we hope to collapse $p(\mathbf{c})$ to a delta distribution at the optimal (MAP) configuration, $\mathbf{c}^*$, because

$$\mathcal{L}_{\mathbf{c}^*} = \mathbb{E}_{\delta_{\mathbf{c}, \mathbf{c}^*}}\left[\mathcal{L}_\mathbf{c}\right] \geq \mathbb{E}_{p(\mathbf{c})}\left[\mathcal{L}_\mathbf{c}\right] = \widetilde{\mathcal{L}}. \tag{17}$$

Effectively, we can search for the dependency structure with the highest ELBO by optimizing the distribution parameters of $p(\mathbf{c})$. Although this optimization procedure may arrive at locally optimal dependency structures, our hope is that these learned structures will still perform better than an arbitrary, predefined dependency structure.

# B    IMPLEMENTATION DETAILS

## B.1    NETWORK ARCHITECTURE

We document the network architectures used in our experiments. We use the same network architectures for all datasets. Input size (pixels per image) is the only difference across datasets. The input_size is $28 \times 28$ for MNIST and Omniglot, and $3 \times 32 \times 32$ for CIFAR-10.

**Encoders:**

```
fc(input_size,512) → batch_norm → ELU → fc(512,512) → batch_norm → ELU →
fc(512,256) → batch_norm → ELU → fc(256,128)
```

**Latent Dependencies:** Latent dependencies are modelled by non-linear MLPs. Note that the top-down architecture is shared between inference model and generative model, but the MLPs are optimized independently.

*bottom-up:* For each node with $N'$ dimensions, the local potential is predicted by a mapping from the encoded feature:

`fc(128,128)` → batch_norm → ELU → `fc(128,N')`. The output feature is then mapped to $\mu$ and $\log$ var with two independent `fc(N',N')` layers, respectively.

*top-down:* For each node ($N'$ dimensions) with a set of parent nodes, the top-down inference/generation is implemented as:

`fc(sum of parent nodes' dimension, 128)` → batch_norm → ELU → `fc(128,N')`. The output feature is then mapped to $\mu$ and $\log$ var with two independent fc($N'$, $N'$) layers, respectively.

**Decoders:**

`fc(N',256)` → batch_norm → ELU → `fc(256,512)` → batch_norm → ELU → `fc(512,512)` → batch_norm → ELU → `fc(512,input_size)` → output_function()

`output_function` for MNIST and Omniglot is `sigmoid()`, which predicts the mean $\mu$ of Bernoulli observations; and `sigmoid()` predicting $\mu$, `fc(input_size,input_size)` predicting $\log$ var of Gaussion observations for CIFAR-10.

## B.2    TRAINING

All models were implemented with PyTorch (Paszke et al., 2017) and trained using the Adam (Kingma & Ba, 2015) optimizer with a mini-batch size of 64 and learning rate of $1e^{-3}$. Learning rate is decresed by 0.25 every 200 epochs. The Gumbel-softmax temperature was initialized at 1 and decreased to $0.99^{\text{epoch}}$ at each epoch. MNIST and Omniglot took 2000 epochs to converge, and CIFAR took 3000 epochs to converge.

## B.3    INFERENCE MODULE DETAILS

Parameter prediction for the local factors $q_\phi(\mathbf{z}_n | \mathbf{x}, \mathbf{z}_{pa(n)})$ consists of a precision-weighted fusion of the *top-down prediction* $\widehat{\psi}_n$ and a *bottom-up prediction* $\widetilde{\psi}_n$. Specifically, for a latent variable $\mathbf{z}_n$, $\widehat{\psi}_n \coloneqq \{\widehat{\boldsymbol{\mu}}_n, \widehat{\boldsymbol{\sigma}}_n\}$, and $\widetilde{\psi}_n \coloneqq \{\widetilde{\boldsymbol{\mu}}_n, \widetilde{\boldsymbol{\sigma}}_n\}$.

*Bottom-up Inference.* A high-dimensional input is first mapped to a feature vector $h_x$ by an encoder MLP. $h_x$ is then used to predict $\widetilde{\boldsymbol{\mu}}_n$ and $\widetilde{\boldsymbol{\sigma}}_n$ with non-linear $\text{MLP}_n^{BU}$.

*Top-down Inference.* $\widehat{\boldsymbol{\mu}}_n$ and $\widehat{\boldsymbol{\sigma}}_n$ are predicted by $\widehat{\boldsymbol{\mu}}_n = \text{MLP}_n^{TD}\left(\left[\mathbf{z}_{pa(n)} \odot \mathbf{c}_{pa(n),n}\right]\right)$ and $\widehat{\boldsymbol{\sigma}}_n = \text{MLP}_n^{TD}\left(\left[\mathbf{z}_{pa(n)} \odot \mathbf{c}_{pa(n),n}\right]\right)$, respectively. [,] denotes concatenation operation, and $\odot$ denotes element-wise multiplication.

*Precision-weighted fusion.* Having $\widehat{\psi}_n$ and $\widetilde{\psi}_n$, the parameters of the local conditional distribution is given by $q_\phi(\mathbf{z}_n | \mathbf{x}, \mathbf{z}_{pa(n),\mathbf{c}_{pa(n),n}}) \sim \mathcal{N}(\mathbf{z}_n | \mu_n, \sigma_n^2)$, with

$$\sigma_n = \frac{1}{\widehat{\sigma}_n^{-2} + \widetilde{\sigma}_n^{-2}},$$
$$\mu_n = \frac{\widehat{\mu}_n \widehat{\sigma}_n^{-2} + \widetilde{\mu}_n \widetilde{\sigma}_n^{-2}}{\widehat{\sigma}_n^{-2} + \widetilde{\sigma}_n^{-2}}. \tag{18}$$

## C  ADDITIONAL RESULTS

### C.1  ROBUSTNESS OF LOG-LIKELIHOODS

In Fig. 5, we report the averaged test log-likelihoods and associated standard deviations of Graph VAE and our baselines at different epochs. All calculations are based on 5 independent runs.

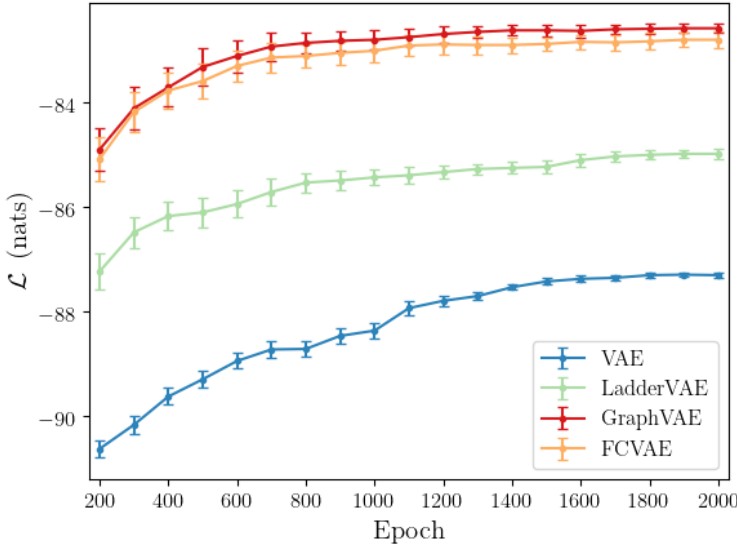

Figure 5: **Comparison of Test-time Log-Likelihoods on MNIST.** Models are trained 5 times individually, and test-time performances are evaluated every 300 epochs, with mean log-likelihood and standard deviation indicated by the colored bars.

### C.2  LATENT EMBEDDINGS

Our training objective optimizes intrinsic structure (which does not necessarily correlate with semantic meaning) and does not incentivize a disentanglement of latent factors. Interestingly, a TSNE-visualization of the data as well as latent embeddings of Graph VAE and VAE on MNIST (Fig. 6) shows that the latent embedding of Graph VAE exhibits a large (semantic) gap between different classes, even though the model is trained in an unsupervised fashion. We will further investigate this behavior in future work.

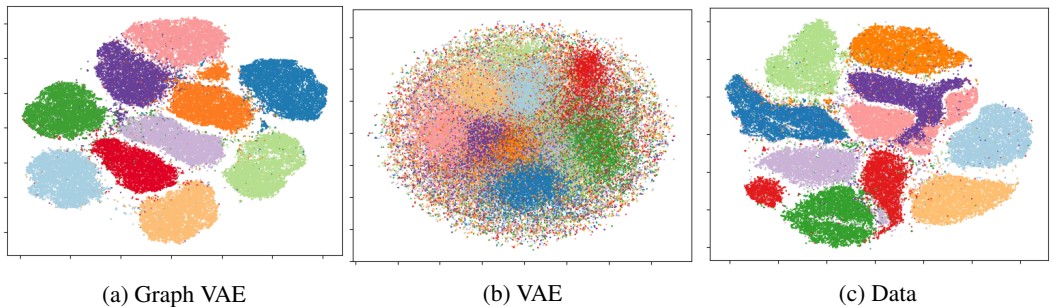

| (a) Graph VAE | (b) VAE | (c) Data |

Figure 6: **TSNE-Visualization of Latent Embeddings on MNIST.** We visualize embeddings of (a) Graph VAE, (b) VAE, and (c) the data itself.

