# OpenReview forum: "Variational Autoencoders with Jointly Optimized Latent Dependency Structure"
_ICLR.cc/2019/Conference_

### Official Review · AnonReviewer2 · 2018-10-22
**Interesting paper but with technical issues that need addressing [now addressed in revision]**

**Rating:** 8
**Confidence:** 5

**Review:**

This paper presents a VAE approach in which a dependency structure on the latent variable is learned during training.  Specifically, a lower-triangular random binary matrix c is introduced, where c_{i,j} = 1 for i>j, indicates that z_i depends on z_j, where z is the latent vector.  Each element of c is separately parametrized by a Bernoulli distribution whose means are optimized for during training, using the target \mathbb{E}_{p(c)}[\mathcal{L}_c] where \mathcal{L}_c indicates the ELBO for a particular instance of c.  The resulting "Graph-VAE" scheme is shown to train models with improved marginal likelihood than a number of baselines for MNIST, Omniglot, and CIFAR-10.

The core concept for this paper is good, the results are impressive, and the paper is, for the most part, easy to follow.  Though I think a lot of people have been thinking about how to learn dependency structures in VAEs, I think this work is the first to clearly lay out a concrete approach for doing so.  I thus think that even though this is not the most novel of papers, it is work which will be of significant interest to the ICLR community.  However, the paper has a number of technical issues and I do not believe the paper is suitable for publication unless they are addressed, or at the vest least acknowledged. I further have some misgivings with the experiments and the explanations of some key elements of the method.  Because of these issues, I think the paper falls below the acceptance threshold in its current form, but I think they could potentially be correctable during the rebuttal period and I will be very happy to substantially increase my score if they are; I feel this has the potential to be a very good paper that I would ultimately like to see published.

%%% Lower bound %%%

My first major concern is in the justification of the final approach (Eq 8), namely using a lower bound argument to move the p(c) term outside of the log.  A target being a lower bound on something we care about is never in itself a justification for that target -- it just says that the resulting estimator is provably negatively biased.  The arguments behind the use of lower bounds in conventional ELBOs are based on much more subtle arguments in terms of the bound becoming tight if we have good posterior approximations and implicit assumptions that the bound will behave similarly to the true marginal.  The bound derived in A.1 of the current paper is instead almost completely useless and serves little purpose other than adding "mathiness" of the type discussed in https://arxiv.org/abs/1807.0334. Eq 8 is not a variational end-to-end target like you claim.  It is never tight and will demonstrably behave very differently to the original target.

To see why it will behave very differently, consider how the original and bound would combine two instances of c for the MNIST experiment, one corresponding to the MAP values of c in the final trained system, the other a value of c that has an ELBO which is, say, 10 nats lower.  Using Eq 8, these will have similar contributions to the overall expectation and so a good network setup (i.e. theta and phi) is one which produces a decent ELBO for both.  Under the original expectation, on the other hand, the MAP value of c corresponds to a setup that has many orders of magnitude higher probability and so the best network setup is the one that does well for the MAP value of c, with the other instance being of little importance.  We thus see that the original target and the lower bound behave very differently for a given p(c).

Thankfully, the target in Eq 8 is a potentially reasonable thing to do in its own right (maybe actually more so that the original formulation), because the averaging over c is somewhat spurious given you are optimizing its mean parameters anyway.  It is easy to show that the "optimum" p(c) for a given (\theta,\phi) is always a delta function on the value of c which has the highest ELBO_c.  As Fig 3 shows, the optimization of the parameters of p(c) practically leads to such a collapse.  This is effectively desirable behavior given the overall aims and so averaging over values of c is from a modeling perspective actually a complete red herring anyway.  It is very much possible that the training procedure represented by Eq 8 is (almost by chance) a good approach in terms of learning the optimal configuration for c, but if this is the case it needs to be presented as such, instead of using the current argument about putting a prior on c and constructing a second lower bound, which is a best dubious and misleading, and at worst complete rubbish.  Ideally, the current explanations would be replaced by a more principled justification, but even just saying you tried Eq 8 and it worked well empirically would be a lot better than what is there at the moment.

%%% Encoder dependency structure does not match the generative model %%%

My second major concern is that the dependency structure used for the encoder is incorrect from the point of view of the generative model.  Namely, a dependency structure on the prior does not induce the same dependency structure on the posterior.  In general, just because z_1 and z_2 are independent, doesn't mean that z_1 and z_2 are independent given x (see e.g. Bishop).  Consequently, the encoder in your setup will be incapable of correctly representing the posterior implied by the generative model.  This has a number of serious practical and theoretical knock-on effects, such as prohibiting the bound becoming tight, causing the encoder to indirectly impact the expressivity of the generative model etc.  Note that this problem is not shared with the Ladder VAE, as there the Markovian dependency structure means produces a special case where the posterior and prior dependency structure is shared.

As shown in https://arxiv.org/abs/1712.00287 (a critical missing reference more generally), it is actually possible to derive the dependency structure of the posterior from that of the prior.  I think in your case their results imply that the encoder needs to be fully connected as the decoder can induce arbitrary dependencies between the latent variables.  I am somewhat surprised that this has not had more of an apparent negative impact on the empirical results and I think at the very very least the paper needs to acknowledge this issue.  I would recommend the authors run experiments using a fully connected encoder and the Graph-VAE decoder (and potentially also vice verse).  Should this approach perform well, it would represent a more principled approach to replace the old on from a generative model perspective.  Should it not, it would provide an empirical justification for what is, in essence, a different restriction to that of the learned prior structure: it is conceivably actually the case that these encoder restrictions induce the desired decoder behavior, but this is distinct to learning a particular dependency structure in the generative model.

%%% Specifics of model and experiments %%%

Though the paper is generally very easy to read, there as some key areas where the explanations are overly terse.  In particular, the explanation surrounding the encoding was difficult to follow and it took me a while to establish exactly what was going on; I am still unsure how \tilde{\psi} and \hat{\psi} are combined.  I think a more careful explanation here and a section giving more detail in the appendices would both help massively.

I was not clear on exactly what was meant by the FC-VAE.  I do not completely agree with the assertion that a standard VAE has independent latents.  Though the typical choice that the prior is N(0,I) obviously causes the prior to have independent latents, as explained earlier, this does not mean the latents are independent in the posterior.  Furthermore, the encoder implicitly incorporates these dependencies through its mean vector, even if it uses a diagonal covariance (which is usually rather small anyway).  What is actually changed from this by the FC-VAE?  Are you doing some kind of normalizing flow approach here?  If so this needs proper explanation.

Relatedly, I am also far from convinced by the arguments presented about why the FC-VAE does worse at the end of the experiments.  VAEs attempt to maximize a marginal likelihood (through a surrogate target) and a model which makes no structural assumptions will generally have a lower marginal likelihood than one which makes the correct structural assumptions.  It is thus perfectly reasonable that when you learn dependency structures, you will get a higher marginal likelihood than if you presume none.  I thus find your arguments about local optima somewhat speculative and further investigation is required.

%%% Experiments %%%

Though certainly not terrible, I felt that the experimental evaluation of the work could have been better.  The biggest issue I have is that no error bars are given for the results, so it is difficult to assess the robustness of the Graph-VAE.  I think it would be good to add convergence plots with error bars to see how the performance varies with time and provide an idea of variability.  More generally, the experiment section overall feels more terse and rushed than the rest of the paper, with some details difficult to find or potentially even straight up missing.

Though Fig 3 is very nice, it would be nice to have additional plots seeing qualitatively what happens with the latent space.  E.g. on average what proportion of the c tend to zero?  Is the same dependency structure always learned?  What do the dataset encodings look like?  Are there noticeable qualitative changes in samples generated from the learned models?  I would be perfectly happy for the paper to extend over the 8 pages to allow more results addressing these questions.

---

> ### Author Response · Authors · 2018-11-20
> **Reply to Reviewer 3**
>
> (1) Lower bound:
>  Thank you for this insight. Indeed, the lower bound induced by the distribution over dependency variables (Eq.(8)) has different properties than the original ELBO. In particular, it is not a proper variational objective and is not guaranteed to be tight if the approximate posterior matches the true posterior. However, as correctly pointed out, this lower bound recovers the ELBO when the distribution over dependency variables converges to a fixed estimate. In practice, we observe this convergence (Fig. 3), and by the end of training, we are effectively optimizing the ELBO for a single dependency structure. Placing a distribution over dependency structures and optimizing $\widetilde{\mathcal{L}}$ instead of $\mathcal{L}$ should therefore be seen as an annealing technique, facilitating optimization over discrete dependency structures, rather than a strict extension of the variational framework. We have modified Section 3 and Appendix A to more clearly communicate this perspective. While we agree that ``mathiness'' should be kept to a minimum, the derivation in Appendix A shows how the dependency variables affect the ELBO, explaining why we expect the model to converge to a fixed structure. We feel this is helpful for understanding the approach and its behaviour during training.
> (2) Encoder Dependency:
> Thank you for pointing this out. We agree that the structure of our encoder does not capture all of the possible dependencies in the true posterior, and we have added this point to Section 3.1 for clarity. However, the same is true for vanilla VAEs, where all latent variables are sampled from the approximate posterior independently, i.e. the distribution is axis-aligned in the latent dimensions. Variational inference does not specify whether an approximate posterior is "correct" or "incorrect"; there are only varying degrees in the quality of approximation. Models are also trained to adapt to their approximate posteriors. Intriguingly, we trained a Graph VAE decoder using a fully-connected encoder and found that the log-likelihood of this model was -83.21 nats, which is closer to the performance of FC VAE. Likewise, the log-likelihood of a fully-connected decoder with a Graph VAE encoder was -83.20 nats. Thus, restricting the approximate posterior dependency structure to mirror the prior seems to improve the final performance. This observation could be explored further in future work.
> (3) Inference module description:
> We have added a detailed description of the inference module in Appendix B.3.
> (4) Clarification on FC VAE:
> Normalizing flows attempts to account for dependencies in the approximate posterior. On the other hand, FC VAEs, and hierarchical VAEs more generally, add dependencies to the prior on the latent variables. This has been shown to improve the flexibility of the model over the diagonal standard Gaussian priors in Vanilla VAEs. Top-down inference (Sønderby, et al., 2016) is one approach for capturing these dependencies in the approximate posterior as well.
> (5) Explanation for FC VAE comparison:
> We have updated the discussion at the end of Section 5 to reflect the fact that, at this point, it is still somewhat speculative that the performance improvements in Graph VAE are due to difficulties in optimization. We agree with the point that learning the structure should result in equal or better performance than assuming a fixed structure. However, FC VAE and Graph VAE are both capable of learning the same set of models, i.e. the same hypothesis space. Graph VAE could conceivably retain all of the dependencies. Likewise, FC VAE could set entire layers of weights to zero, thereby removing dependencies. Thus, it may not be whether structure can be learned, but rather the ease with which it can be learned. Graph VAE can effectively take larger jumps during optimization by modifying a single gating parameter. FC VAE, on the other hand, requires coordinated steps along many parameter dimensions to achieve the same effect. Properly evaluating this perspective will require further follow-up work.
> (6) Robustness:
> Please refer to R2(1) for additional results regarding the robustness of the proposed model, including error bars, a significance test, and stability of the learned structure.
> (7) Semantics:
> Please refer to R1(5) for a perturbation experiment providing additional insight into the encodings of local nodes. Additionally, we have included a visualization of the latent space using a TSNE-embedding in Appendix C.2.

---

> > ### Comment · AnonReviewer2 · 2018-11-21
> > **Significant improvement**
> >
> > Thank you for the follow-up and revision. Overall, I was very happy that the authors have a made a very genuine effort to address my concerns and have increased my score appropriately from a 5 to a 7.
> >
> > The main criticism that I do not think was fully addressed (though there was certainly a large improvement) is that about the encoder dependency structure not matching the decoder.  The quoted results here are extremely interesting and are essential to trying to understand exactly where the improvements are really coming from.  Consequently, I think this point needs a lot more consideration in the paper including rerunning these experiments the same way as in Figure 5 and including them in the paper itself; both values are within the variability of the Graph-VAE so I don't think you can reach the conclusions from your response just yet.  I also think I more careful discussion of this is required.  It is probably not feasible to fully answer everything now, but it would be good to at least think about exactly how the encoder structuring influences the behavior of the generative model and highlight some question for future work on this subject which I think is actually right at the crux of the work.  I might be willing to increase my score further to an 8 if this can be adequately addressed.
> >
> > Other specific comments:
> > 1) I particularly enjoyed the edits made to address my concerns about the lower bound.  I think this is introduced a lot better now and is no longer misleading or guilty of mathsiness.
> > 2) The extra experimentation was much appreciated.  I think there is still scope to improve this further for the final paper, but they are no longer a cause for concern.
> > 3) The values in Table 1 need updating as the LL for the Graph VAE from the original set of results was clearly overestimated given the average performance in the new figure.
> > 4) Though I appreciate this will probably not be feasible to achieve during the revision period, having error bars for all the experiments instead of just MNIST would really improve the impact of the results.  The y axis label for new figure should also be changed as it currently makes it look like the ELBO instead of the LL.
> > 5) I think the FC-VAE still needs a more careful introduction - presumably this is the Graph-VAE with all c set to one?  If so, say this explicitly.
> > 6) I still find the arguments at the end of section 5 a little strong.

---

> > > ### Author Response · Authors · 2018-11-27
> > > **Reply to Reviewer 3**
> > >
> > > We appreciate the positive feedback and are grateful for the helpful insights; they have certainly improved our presentation.
> > >
> > > (encoder/decoder relationship) As suggested, we have computed mean and standard deviation of the experiment with fully-connected encoder graph and learned decoder graph (5 runs). We have added a discussion of these results to section 4.3 and have also included them in Appendix C.1. Furthermore, we have expanded our discussion of the relationship between the structures of the generative model and its corresponding posterior (first paragraph on page 5), which we hope provides a clearer impression of (1) their connection; and (2) the effects resulting from our parameter sharing approach.
> > > (3/4) Computing mean values and standard deviations for the experiments in Table 1 requires training/evaluation of a total of 60 models (5 training runs of 4 models on 3 datasets) w.r.t. 3 performance metrics (LL/ELBO/KL). The majority of these experiments is still running and, for the sake of consistency and to avoid confusion, we decided not to do a partial update of Table 1, which would have resulted in a mix of averaged results and single-run results. An update with error bars for all entries of Table 1, consistent with the supplemental material, will be included in the camera-ready version.
> > > (5) Indeed, we implement the FC-VAE as a VAE with predefined structure $c_{i,j}=1$ for all $i>j$. We have added this information to section 4.3.
> > > (6) We have changed the wording in section 5.3, so that the presented arguments are not perceived too strong.

---

> > > > ### Comment · AnonReviewer2 · 2018-11-27
> > > > **Further improvement**
> > > >
> > > > Thank you for your further updates which I think improve the paper further.  I have consequently decided to increase my score to an 8.  One final question that would be good to answer in the camera ready if you can would be to try and see if you can establish what dependency structure (if any) for the generator is compatible will the learned encoder (i.e. is there any generative model for which the learned encoder structure is a faithful inverse).  This is not a critical point though and not something I would expect to be successfully addressed during the discussion period.

---

### Official Review · AnonReviewer3 · 2018-10-31
**An idea with potential, but weakly developped and tested**

**Rating:** 6
**Confidence:** 3

**Review:**

The authors propose to augment the latent space of a Variational AutoEncoder [1] with an auto-regressive structure, to improve the expressiveness of both the inference network and the latent prior, making them into a general DAG of latent variables. This works goes further in the same direction as the Ladder VAE [2]. This paper introduces a mechanism for the latent model to directly learn its DAG structure by first considering the fully-connected DAG of latent variables, and adding Bernoulli variables controlling the presence or absence of each edge. The authors derive a new ELBO taking these variables into account, and use it to train the model. The gradients of the parameters of the Bernoulli variables are computed using the Gumbel-Softmax approach [3] and annealing the temperature.

The authors observe with they experiments that the Bernoulli variables converge relatively quickly towards 0 or 1 during the training, fixing the structure of the DAG for the rest of the training. They test their model against a VAE, a Ladder VAE and an alternative to their model were the DAG is fixed to remain fully-connected (FC-VAE), and observe improvements in terms of the ELBO values and log-likelihood estimations.

The main addition of this paper is the introduction of the gating mechanism to reduce the latent DAG from its fully-connected state. It is motivated by the tendency of latent models to fall into local optima.

However, it is not clear to me what this mechanism as it is now adds to the model:

- The reported results shows the improvements of Graph-VAE over FC-VAE to be quite small, making their relevance dubious in the absence of measurement of variance accross different trainings. Additionally, the reported performances for Ladder VAE are inferior to what [2] reports. Actually the performance of Ladder-VAE reported in [2] is better than the one reported for Graph-VAE in this paper, both on the MNIST and Omniglot datasets.

- The authors observe that the Bernoulli variables have converged after around ~200 epochs. At this time, according to their reported experimental setup, the Gumbel-Softmax temperature is 0.999^200 ~= 0.82, which is still quite near 1.0, meaning the model is still pretty far from a real Bernoulli-like behavior. And actually, equation 9 is not a proper description of the Gumbel-Softmax as described by [3] : there should be only 2 samples from the Gumbel distribution, not 3. Given these two issues, I can't believe that the c_ij coefficients behave like Bernoulli variables in this experiment. As such, It seems to me that Graph-VAE is nothing more than a special reparametrization of FC-VAE that tends to favor saturating behavior for the c_ij variables.

- On figure 3b, the learned structure is very symmetrical (z2, z3, z4 play an identical role in the final DAG). In my opinion, this begs for the introduction of a regulatory mechanism regarding the gating variable to push the model towards sparsity. I was honestly surprised to see this gating mechanism introduced without anything guiding the convergence of the c_ij variables.

I like the idea of learning a latent structure DAG for VAEs, but this paper introduces a rather weak way to try to achieve this, and the experimental results are not convincing.

[1] https://arxiv.org/abs/1312.6114
[2] https://arxiv.org/abs/1602.02282
[3] https://arxiv.org/abs/1611.01144

---

> ### Author Response · Authors · 2018-11-20
> **Reply to Reviewer 2**
>
> (1) Robustness & comparison with FC VAE:
> We have computed the mean and standard deviation of the test log-likelihood on MNIST across 5 independent runs. Our results are shown in Appendix C, demonstrating a stable learning process. The optimization process is also robust to initialization and other sources of uncertainty and typically converges to the same latent structure. Furthermore, we have addressed the question of significance with respect to FC VAE with a Mann-Whitney U-Test, which rejects the null hypothesis of equal log-likelihood distributions at the 0.01 significance level (p=0.004). We discuss this performance improvement over FC VAE in detail in Section 5.
> (2) Ladder VAE performance:
> Model performance is influenced by a multitude of factors. The focus of this paper is on latent structures and our experiments are designed in way that isolates the effects of a model's latent structure as well as possible. To this end, we use the same encoder/decoder structure (Appendix B) and the same number of latent dimensions (80) in all experiments. While following this principle forced us to deviate from the encoder/decoder structure used in (Sønderby, et al., 2016), resulting in different test log-likelihoods, our experiments constitute a correct and fair evaluation of all models on equal grounds. Please also refer to R1(3) for a discussion of an additional Ladder VAE experiment with varying node dimension.
> (3) Gumbel-softmax:
> We thank the reviewer for pointing out two typos that we have corrected in our revised version: (3-1) We multiply the current temperature by 0.99 after each epoch, not 0.999. The temperature after $200$ epochs is thus approximately 0.13, not 0.82. (3-2) Following code from Categorical VAE (Jang et al., 2017), we use only 2 samples to form the Gumbel-softmax distribution, not 3. We want to emphasize that these typos do not carry over to our code and do not affect any of our experimental results. Our implementation will be made publicly available after the review period.
> (4) Symmetric/sparse structures:
> We have included Fig. 3(b) to give a general impression of the learned latent structure, but not all of its properties generalize to different data or a different training setup. In particular, the observed symmetry does not generalize to another M/N-ratio: With N=8 nodes, which is the number used in Table 1, the learned dependency structure does not exhibit any symmetry. Convergence to a sparse structure is discussed in R3(1) and R3(4).

---

> > ### Comment · AnonReviewer3 · 2018-11-24
> > **Most concerns adressed, performance comparison remains weak**
> >
> > Thanks for your answer corrections, and additions to the paper. This mainly resolves the issues I noted. As such I see your contribution as an alternative parametrization compared to the FC-VAE which changes the optimization landscape, favoring more sparse latent graphs. This is worth of interest, and I'm raising my rating to reflect that.
> >
> > The comparison of performance however remains fragile in my opinion. Indeed, it is completely relevant to compare models changing only a single thing (here the latent structure and way it is trained). However different factors of change do not interact in ways that are easy to predict, and it is not obvious to me that applying the same change to the original LadderVAE would necessarily similarly improve its performance.

---

### Official Review · AnonReviewer1 · 2018-11-03
**Interesting idea: use a matrix of binary random variables to capture dependencies between latent variables in a hierarchical deep generative model.**

**Rating:** 7
**Confidence:** 4

**Review:**

Often in a deep generative model with multiple latent variables, the structure amongst the latent variables is pre-specified before parameter estimation. This work aims to learn the structure as part of the parameters. To do so, this work represents all possible dependencies amongst the latent random variables via a learned binary adjacency matrix, c, where a 1 denotes each parent child relationship.

Each setting of c defines a latent variable as the root and subsequent parent-child relationships amongst the others. To be able to support (up to N-1) parents, the paper proposes a neural architecture where the sample from each parent is multiplied by the corresponding value of c_ij (0'd out if the edge does not exist in c), concatenated and fed into an MLP that predicts a distribution over the child node. The inference network shares parameters with the generative model (as in Sonderby et. al). Given any setting of c, one can define the variational lower-bound of the data. This work performs parameter estimation by sampling c and then performing gradient ascent on the resulting lower-bound.

The model is evaluated on MNIST, Omniglot and CIFAR where it is found to outperform a VAE with a single latent variable (with the same number of latent dimensions as the proposed graphVAE), the LadderVAE and the FCVAE (VAE with a fully connected graph). An ablation study is conducted to study the effect of number of nodes and their dimensionality.

Overall, the paper is (a) well written, (b) proposes a new, interesting idea and (c) shows that the choice to parameterize structure via the use of auxillary random variables improves the quality of results on some standard benchmarks.

Comments and questions for the authors:
* Clarity
It might be instructive to describe in detail how the inference network is structured for different settings of c (for example, via a scenario with three latent variables) rather than via reference to Sonderby et. al.

What prior distribution was used for c?

For the baseline comparing to the LadderVAE, what dimensionalities were used for the latent variables in the LadderVAE (which has a chain structured dependence amongst its latent variables)? The experimental setup keeps fixed the latent dimensionality to 80 -- the original paper recommends a different dimensionality for each latent variables in the chain [https://arxiv.org/pdf/1602.02282.pdf, Table 2] -- was this tried? Did the ladderVAE do better if each latent variable in the chain was allowed to have a dimensionality?

* Related work
There is related work which leverages Bayesian non-parametric models to learn hierarchical priors for deep generative models. It is worth discussing for putting this line of work into context. For example:
http://openaccess.thecvf.com/content_ICCV_2017/papers/Goyal_Nonparametric_Variational_Auto-Encoders_ICCV_2017_paper.pdf
and more recently: https://arxiv.org/pdf/1810.06891.pdf

In the context of defining inference networks for generative models where the latent variables have structure, Webb et. al [https://arxiv.org/abs/1712.00287] describe how inference networks should be setup in order to invert the generative process.

* Qualitative study
Notable in its absence is a qualitative analysis of what happens to the data sampled from the model when the various nodes in the learned hierarchy are perturbed holding fixed their parents. Have you attempted this experiment? Are the edge relationships sensible or interesting?

Is there a relationship between the complexity of each conditional distribution in the generative model and the learned latent structure? Specifically, have you experimented to see what happens to the learned structure amongst the latent variables if each conditional density is a linear function of its parents?

---

> ### Author Response · Authors · 2018-11-20
> **Reply to Reviewer 1**
>
> (1) Inference module description:
> We have added a detailed description of the inference module in Appendix B.3.
> (2) Prior distribution p(c):
> The structure gating variables \textbf{c} follow a Bernoulli distribution with parameters $\mathbf{\mu}$, which are initialized at 0.5. We optimize these parameters during the training process across all data examples.
> (3) Ladder VAE with varying node dimension:
> Following the architecture in the original Ladder VAE paper (Sønderby, et al., 2016), we ran an additional experiment using Ladder VAE with node dimensions (4-8-16-32-64). The test log-likelihood of this model on MNIST is -84.0 nats, which is higher than the version with constant node dimension (-84.8 nats) but lower than FC VAE (-83.0 nats) and the proposed Graph VAE (-82.1 nats). In our experiments, we keep the number of node dimensions constant to guarantee the same total latent dimensionality (80) in all models.
> (4) References:
> We thank the reviewer for pointing out two missing references as well as a recent work that has appeared after our submission. We have added them to our discussion in Section 2.2.
> (5) Node perturbations:
> We did not observe a clear semantic pattern when perturbing a child node while keeping its parents fixed. This is not surprising, because intrinsic structure does not necessarily correlate with semantic meaning and our training objective does not incentivize a semantic disentanglement of latent factors. Instead, we observed that nodes close to the root modulate global, low-frequency structural changes, whereas leaf nodes encode local, high-frequency elements of the data.
> (6) Linear conditional densities:
> Learning linear conditional densities leads to an interesting effect: while the test log-likelihood on MNIST decreases by 2.0 nats to -84.1 nats, the structure now converges to a fully-connected graph. We interpret this behavior as an attempt of the optimization process to compensate less expressive local distributions with additional dependencies.

---

> > ### Comment · AnonReviewer1 · 2018-11-28
> > **Update to author rebuttal**
> >
> > Thank you for the answers to questions regarding the LadderVAE and the choice of prior distribution. The outcome of both the node perturbations and the linear conditional density experiment is in line with I expected. Overall, I the contribution of the work is bit clearer now and I'm raising my score to reflect this.
> >
> > The primary actionable recommendation I have is to add visualizations of the data under different node perturbations to the supplementary material for the best model learned on MNIST/Omniglot so that readers may have a better intuition for what low-frequency structural changes and high-frequency elements correspond to.

---

### Author Response · Authors · 2018-11-20
**Reply to Reviewers**

We want to thank all reviewers for their thorough and valuable feedback. Questions and concerns are discussed below and clarified in the updated paper.

---

### Meta-Review · Area_Chair1 · 2018-12-12
**Slightly hacky but still good progress**

**Confidence:** 4
**Recommendation:** Accept (Poster)

**Metareview:**

Strengths:
This paper develops a method for learning the structure of discrete latent variables in a VAE.  The overall approach is well-explained and reasonable.

Weaknesses:
Ultimately, this is done using the usual style of discrete relaxations, which come with tradeoffs and inconsistencies.

Consensus:
The reviewers all agreed that the paper is above the bar.